# Real-World Performance of a Self-Operated Home Monitoring System for Early Detection of Neovascular Age-Related Macular Degeneration

**DOI:** 10.3390/jcm10071355

**Published:** 2021-03-25

**Authors:** Allen C. Ho, Jeffrey S. Heier, Nancy M. Holekamp, Richard A. Garfinkel, Byron Ladd, Carl C. Awh, Rishi P. Singh, George E. Sanborn, Jennifer H. Jacobs, Michael J. Elman, Anat Loewenstein, David A. Eichenbaum

**Affiliations:** 1Wills Eye Hospital, 840 Walnut St., Philadelphia, PA 19107, USA; achomd@gmail.com; 2Ophthalmic Consultants of Boston, 50 Staniford St., Ste. 600, Boston, MA 02114, USA; jsheier@eyeboston.com; 3Pepose Vision Institute, 1815 Clarkson Road, Chesterfield, MO 63124, USA; nholekamp@gmail.com; 4The Retina Group of Washington, 110 Irving St NW, Washington, DC 20010, USA; rgarfinkel@rgw.com; 5Virginia Eye Institute, 6946 Forest Ave Suite 100, Richmond, VA 23230, USA; laddb@vaeye.com; 6Tennessee Retina, 345 23rd Avenue North, Suite 350, Nashville, TN 37203, USA; carlawh@gmail.com; 7Center for Ophthalmic Bioinformatics, Cole Eye Institute, Cleveland Clinic Foundation, 9500 Euclid Avenue, i-32, Cleveland, OH 44106, USA; singhr@ccf.org; 8Notal Vision, 7717 Coppermine Dr., Manassas, VA 20109, USA; georges@notalvision.com (G.E.S.); jjacobs@notalvision.com (J.H.J.); 9Elman Retina, 7671 Quarterfield Rd #100, Glen Burnie, MD 21061, USA; elman@elmanretina.com; 10Department of Ophthalmology, Tel Aviv Medical Center, Tel Aviv 6209105, Israel; 11The Sackler Faculty of Medicine, Tel Aviv University, Tel Aviv 6997801, Israel; 12Retina Vitreous Associates of Florida, 4344 Central Ave, St. Petersburg, FL 33711, USA; deichenbaum@rvaf.com

**Keywords:** ForeseeHome AMD Monitoring System^®^, neovascular age-related macular degeneration, AREDS2-HOME study

## Abstract

The real-world performance of a home telemonitoring strategy (ForeseeHome AMD Monitoring System^®^, Notal Vision, Inc.,Manassas VA, USA) was evaluated and compared to the device arm of the AREDS2-HOME study among patients with intermediate AMD (iAMD) who converted to neovascular AMD (nAMD). All patients with confirmed conversion to nAMD who used the home monitoring system from 10/2009 through 9/2018 were identified by Notal Vision Diagnostic Clinic’s medical records. Selected outcome variables were evaluated, including visual acuity (VA) at baseline and at conversion, and change in visual acuity (VA) from baseline to time of conversion. In total, 8991 patients performed 3,200,999 tests at a frequency of 5.6 ± 3.2 times/week. The 306 eyes that converted from iAMD to nAMD over the study period (a 2.7% annual rate) were included in the analyses. There was a median (interquartile range) change of −3.0 (0.0–(−10.0)) letters among converted eyes, 81% [95% confidence interval (72–88%)] maintained a VA ≥ 20/40 at the time of conversion, while 69% of the conversion detections were triggered by system alerts. The real-world performance of an at-home testing strategy was similar to that reported for the device arm of the AREDS2-HOME study. The home telemonitoring system can markedly increase early detection of conversion to nAMD.

## 1. Introduction

Eyes with early and intermediate age-related macular degeneration (AMD) typically retain good central vision and overall vision [1]. Any subsequent vision loss is usually attributed to advanced AMD, specifically, geographic atrophy or neovascular AMD (nAMD). Both the Comparison of AMD Treatments Trial (CATT) [2,3] and a real-world analysis that employed the Intelligent Research in Sight (IRIS) registry [4] confirmed that baseline visual acuity (VA) at the time of conversion from AMD to nAMD is a strong predictor of long-term visual outcomes under treatment with intravitreal anti-vascular endothelial growth factor (anti-VEGF) therapies. Significant visual improvement with these treatments, as shown in the landmark registration studies [5,6,7], is often not replicated in real-world outcome studies. Given that baseline vision is a strong determinant of long-term visual outcome and that long-term gains in vision are often unsustainable, early detection of nAMD can be pivotal to achieving favorable therapeutic results [8,9,10,11]. A recent cost-utility analysis reported that early treatment was 138–149% more cost-effective than late treatment, and those authors concluded that early treatment is critical for obtaining optimal vision and cost-effectiveness, as is long-term follow-up and adherence to treatment [12].

Despite the obvious benefit of detecting conversion to nAMD while vision is still good, many reports reveal that only 13 to 35% of patients are diagnosed and begin intravitreal anti-VEGF treatment when their VA is 20/40 or better [4,13,14,15]. One analysis of IRIS registry data on 13,859 subjects found that the mean VA at conversion to nAMD was 20/74 [4]. In another and larger IRIS database analysis, the mean VA of more than 140,000 newly converted eyes was 20/83 [14,15]. Similar findings of poor VA at conversion have been widely reported in other studies in which VA at the time of conversion ranged from 20/80 to 20/142 [16,17,18,19,20,21,22].

The incidence of conversion from dry to nAMD [13,14,23,24,25] suggests that frequent monitoring of eyes with intermediate AMD would be optimal. However, bringing a large, older population into clinics for monitoring visual and anatomic changes can be difficult for both the patient and the caregivers, particularly given current safety concerns related to the COVID-19 pandemic. One possible solution to address these difficulties is at-home self–monitoring. Patients and physicians have long used the Amsler grid as a home monitoring strategy, but it has limited efficacy due to factors, such as problems in filling in the information, lack of fixation, and poor compliance [26,27]. No well-designed clinical trial has shown that the Amsler grid is effective in detecting nAMD in patients with good VA. More effective types of home monitoring technologies and services may well have increasing relevance and desirability in a COVID-19 environment and beyond, as people may prefer to “stay at home” to limit their potential exposure to the virus [28].

The AREDS2-HOME study [29,30] evaluated one home self-monitoring strategy that included the ForeseeHome AMD monitoring system (Notal Vision, Inc., Manassas, VA, USA). The system uses preferential hyperacuity perimetry to detect minute differences in the relative spatial localization of two or more objects and was previously described [1]. In brief, during the test of each eye, the patient is responding to fast stimuli in random locations in the visual field of the central 14°. These flashing signals include artificial distortions with varying amplitudes. The marking of the patient in the location of the presented distortion, in a different location, or the absence of a response are collected, transmitted to the secured cloud location, and are analyzed by the system’s artificial intelligence algorithm. Upon identification of a statistically significant change in the testing results compared to a baseline period, a change alert is communicated to the prescribing physician through a remote diagnostic clinic that provides the monitoring service. The study randomized subjects to either a device arm that included the home monitoring system in addition to standard care or to a control arm that included standard care alone. The study findings showed that a significantly higher percentage of patients were able to maintain VA of 20/40 or better after using the strategy that included home monitoring to help identify conversion in early stages.

The objective of this study was to evaluate the performance in real-world implementation of the strategy that includes the use of this at-home monitoring system in combination with standard care, i.e., the strategy implemented in the device arm of the HOME study. The system allows the patients to proactively monitor their visual status between doctor visits while at the same time establishing a safety net that is intended to prevent significant loss of vision during the conversion to nAMD. This evaluation focused on the most relevant parameters of vision preservation. We hypothesized that since the real-world implementation uses the same device, hyperacuity test, remote diagnostic clinic infrastructure, and compliance reminder services all integrated into a monitoring program, then the efficacy of the system would be similar to the efficacy reported by the participants randomized to the device arm of the AREDS2-HOME study.

## 2. Patients and Methods

This is a retrospective review of data on all patients with available information on a confirmed conversion from intermediate AMD to nAMD during their participation in an at-home monitoring program any time during the period from October 2009 through September 2018. The inclusion criteria for enrollment into the program was diagnosis of intermediate dry AMD and best corrected visual acuity of 20/60 or better in any eye that was prescribed. Participation was defined as having a device at home and the availability of a valid baseline. Patients were identified from the medical records of the Notal Vision Diagnostic Clinic (Notal Vision Inc.; Manassas, VA USA), an independent diagnostic testing facility and medical provider of the home monitoring program. Patients were referred to the program by an eyecare professional. Upon referral, the NVDC contacted the patient over the telephone, provided explanations about the nature of the disease and the purposes of the device, and shipped the device to the patient’s home. Following remote training on device operation, the patients established a baseline of testing results. They were instructed and encouraged to self-test daily. The NVDC monitored their compliance to these instructions and reminded patients to self-test when necessary. Artificial intelligence compared the patient’s baseline and most recent sequence of testing results to a normative database and transmitted the information to the NVDC. If there had been a statistically significant change from baseline in recent self-test scores (an alert), a notification was sent to the NVDC. After the alert was reviewed by the NVDC, it was relayed to the referring physician’s office staff who contacted the patient to schedule a clinical evaluation. Other complementary triggers for an in-office visit were routine and symptoms visits. All these visit-triggering modalities jointly comprise the home monitoring strategy under evaluation.

This study was granted an exemption of oversight by an independent institutional review board (IntegReview, Austin, TX, USA) on the basis of Code of Federal Regulations Title 45, Part 46.104. It does not contain any human participants or use of animals.

### 2.1. Outcome Variables

The datasets of eyes with reported conversion during the study period included demographics, overall duration of participation in the monitoring program (as reported in monitoring years as the sum of periods from the first test to the reported conversion of all the study eyes), the number of tests performed during the study period and the mean weekly frequency of testing, the annual observed conversion rates, and the dates of identification of conversion to nAMD confirmed by the treating physicians. They also included the VA at baseline and confirmed conversion (when available) as measured at the physician’s office, and the modality that triggered the in-office visit in which the diagnosis was confirmed following a system alert (as opposed to detection of conversion during routine or symptom-driven visits).

For the purposes of this study, baseline was defined as the point in time when the patient was first prescribed the at-home monitoring system. The testing baseline used by the system in the ongoing monitoring may have been established few days or weeks later due to the delay from enrollment at the clinic to device delivery, setup, and first usage. Outcome measures included the time from baseline to the reported conversion, the proportion of conversions to nAMD detected following an alert by the at-home monitoring system vs. conversions reported as having been identified during an office visit, the VA at baseline and at the time of conversion, the numerical change in VA from baseline to conversion, the percentage of eyes with a VA ≥ 20/40 at time of conversion, and the percentage of eyes with a VA ≥ 20/40 at baseline that maintained a VA ≥ 20/40 at the time of conversion to nAMD. Conversion was confirmed based on the evaluation of nAMD by a retina specialist.

### 2.2. Statistical Analyses

Statistical analyses were primarily limited to descriptive statistics with comparison between sub-cohorts when applicable. VA was reported as Snellen equivalent and converted to Early Treatment Diabetic Retinopathy Study (ETDRS) letters for comparison when applicable and reported in letter changes or Snellen [31].

## 3. Results

During the study period, a total of 8991 patients were enrolled in the program and tested a total of 13,930 eyes, representing an average of 1.55 eyes per patient. They performed 3,200,999 tests during 11,525 years of monitoring. A total of 306 eyes were reported to have converted from intermediate AMD to nAMD during the study period representing a conversion rate of 2.7% per year, and they were included in the current analyses. The mean (±standard deviation, SD) weekly frequency of testing per eye and per patient were 3.7 ± 1.9 and 5.6 ± 3.2, respectively (Figure 1). The patients’ mean age was 75 ± 7.1 years, and 199 (65%) were females (Table 1). Of the 306 eyes with confirmed disease progression, 211 (69%) were identified following the at-home system alert, and the remaining 95 (31%) were identified during a routine or a symptom-driven visit. The duration from time of baseline to conversion to nAMD by the alerting modality is shown in Table 1. The median 5.7-month difference in the time from baseline to conversion between the modalities was not significant (*p* = 0.082), although it did show a trend towards shorter duration for the system alerts.

Table 2 provides details about the VA for the entire cohort and by detection modality. The data included the VA at baseline, at conversion to nAMD, and the change in VA from baseline. Since this report is based on real-world, retrospectively collected data, not all VA values from both time points were available. There was a statistically significant difference in visual acuity between baseline and at the time of conversion (*p* = 0.00014). The mean (SD) and median (interquartile range, IQR) change in VA from baseline to conversion for the 121 eyes with data at both time points was −5.4 (10.0) and −3.0 [0.0–(−10.0)], respectively. For the 95 eyes with VA data that was triggered by system alert, the change in VA from baseline to conversion was a median (IQR) of −2 [0.0–(−10.0)] letters. In comparison, the change in VA from baseline to conversion in the 26 eyes where the conversion was detected during routine or symptom-driven office visits was a median (IQR) of −4.5 [0.0–(−16.5] letters. The difference of 2.5 letters in median change of VA was not statistically significant (*p* = 0.19).

Of the 193 eyes with information about the VA at conversion to nAMD, the VA of 144 eyes (75%; 95% confidence interval, CI [68–81%]) was equal or better than 20/40. Of the 109 eyes with a baseline VA ≥ 20/40, 88 eyes (81%, 95% CI [72–88%]) maintained a VA ≥ 20/40 at the time of conversion and initiation of treatment. Detection was triggered by a system alert in 86 eyes with a baseline VA ≥ 20/40 that had information about VA at the time of conversion, and 71 of those eyes (83%, 95% CI [73–90%]) maintained a VA ≥ 20/40. Seventeen of the 23 eyes with a baseline VA ≥ 20/40 74% (95% CI [52–90%]) in which conversion was detected during a routine or symptom-driven visit maintained a VA ≥ 20/40. There was no significant difference between the alerting modalities and the proportion of eyes with a VA ≥ 20/40 (*p* = 0.38).

## 4. Discussion

This study reports on the real-world performance of a monitoring strategy that included an at-home self-operated monitoring system in conjunction with office visits (both routine and symptom-driven) for early detection of nAMD. This same strategy was evaluated in a prospective multicenter study that found patients in the home monitoring arm lost significantly less VA from baseline to the time of conversion than those being followed by standard care alone [29].

Our current study included a larger group of eyes (*n* = 306) that converted to nAMD compared to the AREDS2-HOME study (*n* = 82). Our reported conversion rate was 2.7% per monitoring year, which is typical for an intermediate AMD population [32,33], and provides an indication that a significant portion of conversions within this cohort was captured during the study. The mean (SD) weekly frequency of testing per eye was 3.7 ± 1.9, which is consistent with 3.44 ± 1.86 tests per week reported in a recent publication on real-world use of the ForeseeHome device [34]. The patients in our study tested themselves 5.6 ± 3.2 times per week, consistent with the frequency of 5.9 tests per week as calculated from a recent real-world report, [34] and more than the frequency of 4.4 ± 1.7 tests per week reported in the AREDS2-HOME study. A comparison of the distribution of detection between the triggering modalities revealed a similar rate of “first to trigger” by the home monitoring system was reported in both studies: it was 69% in the current study compared with 64% in the AREDS2-HOME study in a comparable per protocol (PP1) population of patients who were using the device at the time of nAMD detection, regardless of frequency of use [29]. Our findings suggest that the inclusion of the home monitoring program in an overall monitoring strategy allows the at-home system to detect the majority of disease conversions, thereby providing an effective safety net for patient vision. The median (IQR) change in VA for the entire cohort was −3.0 [0.0–(−10.0)] letters. In comparison, the median (IQR) change in VA for the device arm of the HOME study was very similar, at −3.0 [−1.0–(−10.0)] letters [29].

Our study demonstrated that 81% [95% CI (72–88%)] of eyes with a VA ≥ 20/40 at baseline retained that vision at the time of conversion, which was lower than the 91% reported for a comparable group in the device arm of the HOME study, but higher than the 62% reported in the control arm of the HOME study [29]. The current report also showed a much higher percentage of eyes with a VA of ≥20/40 (75%) at the time of conversion to nAMD when all eyes were considered, other real-world studies of newly diagnosed nAMD reported a percentage of eyes with a VA of 20/40 or better at the time of conversion that ranged from 13.1 to 34.3%, [14,16,35,36] notably including the IRIS registry dataset (*n* = 55,930 eyes), in which 34.3% of eyes had a VA ≥ 20/40 at the time of diagnosis [14]. Caution is recommended when interpreting these comparative data, however, since the population that elects to use a home monitoring technology does not necessarily represent the general population included in the IRIS database. Our current study showed that for this population, which is similar to the PP1 population in the AREDS2-HOME study, there was real-world efficacy in identifying patients with a better vision at the time of conversion through the use of an at-home monitoring system, a finding comparable to the results of the AREDS2-HOME study.

Overall, there was no significant difference between the alerting modalities in terms of the VA in eyes with good vision (>20/40) at the time of conversion in this population. One possible explanation for the different outcomes of the current and IRIS studies may be that the HOME study baseline VA was 83 letters on the ETDRS chart while it was 79 letters in the current study, shifting the distribution of VAs below the threshold of 20/40. Another possible explanation is that the group of patients who converted to nAMD in this paper who elected to use a home-monitoring technology were more sensitive to small subjective changes in vision, than the IRIS population.

The main limitation of this study is that it is a retrospective database analysis. The VA was not available at baseline or at conversion for all patients, and some of the VA values were reported over the telephone, which may have introduced some bias. This review only included patients who converted to nAMD after establishing a baseline on the ForeseeHome monitoring system. In trying to evaluate the sensitivity of the findings to possible methodological limitations resulting from the retrospective, real-world study design, we considered a worst-case scenario in which an unbalanced, larger number of events were triggered by routine visits or symptoms that were not identified and therefore not included in the report, and we found the expected VA loss to be 4.5 letters. Another limitation is that we did not collect information about the outcomes of alerts issued by the home monitoring system that did not result in an immediate identification of conversion to nAMD. These may include alerts that led to conversions to nAMD that were identified after some delay, alerts that led to the diagnosis of non-nAMD pathologies, and those that wound up being false alerts. In some instances, the patient re-established a baseline and resumed testing. However, those limitations were mitigated by study strengths, which include the length of review (9 years) and the very large number of tests and monitoring years. Also, the same system was used in both the real-world and the AREDS2-HOME study, with the same support and reminders as reflected in the high weekly frequency of use. This allowed us to avoid some of the typical difference between studies and real-world evidence. Other strengths were the uniquely large number of newly diagnosed nAMD events that were detected early after a period of close follow-up with the ForeseeHome system, which differs from how a typical cohort of treatment-naïve nAMD eyes are traditionally diagnosed, treated, and reported during an office visit.

Our conclusions differ from those of another recently published real-world data set, [36] primarily because the reports evaluated two different populations of real-world patients. The current study evaluated a patient population that had a device with a valid baseline up to and including the time of conversion. Yu et al. [34] evaluated all patients who were prescribed the device, and they cite a large number of patients who did not or could not establish or re-establish a baseline, as well as a considerable number of patients who discontinued use of the device.

In conclusion, our current study suggests that the consistent long-term use of an at-home monitoring system may provide a significant benefit to patients as a means of increasing early detection of nAMD with good vision, a known strong predictor for long-term preservation of vision [13,14]. AMD has been long established as a common condition in the elderly population [37]. In today’s COVID-19 pandemic environment and beyond, the ability to provide quality care to patients at high risk of converting to nAMD while limiting their potential exposure to COVID-19 bestows additional value. This study provides evidence to demonstrate that an at-home monitoring system can achieve those goals. It may drive physicians, patients, and their families to request access, gain benefit from the expected performance, and drive a large-scale improvement in the quality of life of this growing population.

## Figures and Tables

**Figure 1 jcm-10-01355-f001:**
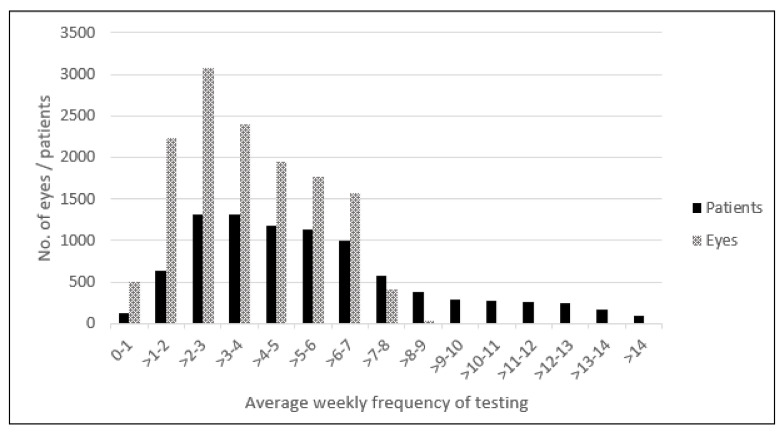
Distribution of the average weekly frequency of testing per eye and per patient.

**Table 1 jcm-10-01355-t001:** Demographics and time to conversion from AMD to nAMD of patients in the at-home monitoring program by modality of detection.

Modality of Detection, *n* (%) of Conversions	All Conversions: 306 (100%)	Conversion Detected Following an Alert: 211 (69%)	Conversion Detected during Routine or Symptom-driven Visit to Physician 95 (31%)
Mean age in years (SD)	75 (7.1)	76 (6.9)	73 (7.3)
Female sex, *n*	199 (65%)	139 (66%)	60 (63%)
Mean (SD) time from initiation of the at-home device use to conversion, months	16.8 (16.3)	15.8 (15.4)	19.2 (18.0)
Median (IQR) time from initiation of the at-home device use to conversion, months	11.4 (4.3–23.5)	10.5 (3.9–22.3)	16.2 (5.4–24.2)

AMD: age-related macular degeneration; nAMD: neovascular age-related macular degeneration; SD: standard deviation; IQR: interquartile range.

**Table 2 jcm-10-01355-t002:** Visual acuity outcomes by detection modality.

VA Outcomes	Baseline VA	VA at Conversion to nAMD	VA at Conversion with Known Baseline VA	VA Change from Baseline to Conversion
No. of eyes	121	193	121	
Mean VA [SD] letters	77.7 (7.4)	72.4 (12.0)	72.3 (12.7)	−5.46 (10.0)
Mean VA Snellen	20/32	20/40	20/40	
Median VA [IQR] letters	79.0 (74.0–84.0)	75.0 (68.0–81.0)	74.0 (66.0–81.0)	−3.0 (0.0–(−10.0))
Median VA Snellen	20/25	20/32	20/32	
Eyes with nAMD detection triggered by a system alert with known VA				
No. of eyes	95	151	95	
Mean VA [SD] letters	77.6 (7.1)	72.7 (11.2)	73.1 (11.3)	−4.5 (8.6)
Mean VA, Snellen	20/32	20/40	20/40	
Median VA [IQR] letters	78.0 (74.0–83.0)	75.0 (69.0–81.0)	74.0 (68.5–81.0)	−2.0 (0.0–(−10.0))
Median VA Snellen	20/32	20/32	20/32	
Eyes with nAMD detection during a routine or symptom-driven visit to physician with known VA				
No. of eyes	26	42	26	
Mean VA [SD] letters	78.0 [8.7]	71.3 [14.7]	69.4 [16.8]	−8.6 [13.9]
Mean VA Snellen	20/32	20/40	20/40	
Median VA [IQR] letters	81.0 [71.75–85.75]	76.0 [62.25–81.0]	74.0 [56.25–82.5]	−4.5 [0.0–(−16.5]
Median VA Snellen	20/25	20/32	20/32	

VA; visual acuity; AMD: age-related macular degeneration; nAMD: neovascular age-related macular degeneration; SD: standard deviation; IQR: interquartile range.

## Data Availability

The datasets generated and analyzed during the current study are not publicly available since the data could be used to derive business intelligence information about the company that owns the clinical data and considers them confidential.

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
