# Peer review of "Real-World Performance of a Self-Operated Home Monitoring System for Early Detection of Neovascular Age-Related Macular Degeneration"

_jcm, 2021, doi:10.3390/jcm10071355_

Round 1
Reviewer 1 Report
The paper reported the real-world performance of a self-operated home monitoring system of early detection of neovascular AMD. The paper providing some interesting data. However, the manuscript can be improved as following:
- Please clarify the baseline visual acuity and visual acuity at conversion are based on the visual acuity self-tested using the home monitoring system, or is based on the visual acuity measured at the physician office visit?
- For Table 2, there are a column for “VA at conversion to nAMD” and a column for “VA at conversion”? What are their difference?
- In Table 1, there are total of 306 eyes with conversion, but in the Table 2, there are only 193 eyes were reported for their visual acuity, why these two numbers are so different? Are there some eyes not having visual acuity measured at baseline or at the time of conversion? This should be made clear in the Results section.
- In Table 2, could you provide p-value for testing whether there is any statistically significant difference in visual acuity between baseline and at the time of conversion?
- In the results, could you provide how many alarms occurred among total of 3,200,999 tests using home monitoring system, and how many are correct alarm and how many are false alarm?
- Is there any information on the CNV lesion size or severity of nAMD at the time of conversion? Is the CNV lesion size different in eyes alarmed from home monitoring system and from the doctor office visit?
- What is the time interval between alarm from home monitoring system to subsequent occurrence of the office visit?
Author Response
The paper reported the real-world performance of a self-operated home monitoring system of early detection of neovascular AMD. The paper providing some interesting data. However, the manuscript can be improved as following:
Comment 1: Please clarify the baseline visual acuity and visual acuity at conversion are based on the visual acuity self-tested using the home monitoring system, or is based on the visual acuity measured at the physician office visit?
Clarification: We accept the comment and have added a clarification. Change: We have updated the Methods section. “They also included the VA at baseline and confirmed conversion (when available) as measured at the physician office.”
Comment 2: For Table 2, there are a column for “VA at conversion to nAMD” and a column for “VA at conversion”? What are their difference?
Clarification: Many thanks indeed for this comment. It was an omitting during the writing process. The values represented in that column allow to calculate the change in VA. It is now corrected. Change: We have updated the table and the title now is “VA at Conversion with known baseline VA.”

Reviewer 2 Report
Thank you for asking me to review this article. It addresses the need for home monitoring of intermediate AMD patients with an attempt to identify conversion to nAMD earlier and thus result in better visual outcomes after treatment. It is a retrospective database study looking at real world performance of at-home monitoring.
The article is well written, with clear methodology and conclusions. It clearly states and expands on its own limitations.
My overall view, as a retinal specialist, is that this paper is probably not best published in a general medical journal such as this, because the likely interest to the average reader or non-retinal specialist is low.
I would suggest you aim for a retinal journal.
Author Response
Reviewer 2
Thank you for asking me to review this article. It addresses the need for home monitoring of intermediate AMD patients with an attempt to identify conversion to nAMD earlier and thus result in better visual outcomes after treatment. It is a retrospective database study looking at real world performance of at-home monitoring.
The article is well written, with clear methodology and conclusions. It clearly states and expands on its own limitations.
Comment 1: My overall view, as a retinal specialist, is that this paper is probably not best published in a general medical journal such as this, because the likely interest to the average reader or non-retinal specialist is low. I would suggest you aim for a retinal journal.
Response 1: Thank you for the comment however, we respectfully disagree. Dry AMD is a very common disease of the growing elderly population, therefore often managed by “non-retina specialists” eye care professional and in some communities maybe even by general practitioners. Therefore, we are in position that this special issue "Diagnosis, Treatment and Prevention of Age-Related Macular Degeneration” can serve as a proper venue to report about a new home-based telemedicine approach to a wider audience.

Reviewer 3 Report
General comments
I am very glad the authors wrote this paper. It is an accurately, needed and useful for the detection of the progression of AMD. This is a well written paper demonstrating efficacy of visual acuity control in the advancement of AMD.
Specific comments for revision:
Introduction
They should describe in more detail the development of the telemonitoring strategy. Could benefit from some further explanation for clarity.
They must include the author (Zapata et al., 2021)
Materials and Methods
The study deals with the measurement of visual acuity thus, the methodically crucial factor.
In the methods, a number of unclear circumstances can be found which have to be carefully clarified.
For example, type of test in the measurement, light conditions of the room, visual acuity was measured monocularly and binocularly.
What were the inclusion and exclusion criteria?
We have previous evidence suggesting that refractive error may be associated with AMD.
Was the refractive error taken into account?
Diabetes is a risk factor for AMD, stronger for late AMD than for earlier stages.
Was it taken into account whether the patients included were diabetic?
Results
The tables present the data nicely but a bit more explanation of the content and implications of the data in the tables would be beneficial.
Discussion
The discussion of data citation was good, but I would have liked to see some views on a way forward.
Finally, is that there is what is the practical impact of this study?
Zapata, M. A., Burés, A., Gallego-Pinazo, R., Gutiérrez-Sánchez, E., Oléñik, A., Pastor, S., … Abraldes, M. (2021). Prevalence of age-related macular degeneration among optometric telemedicine users in Spain: a retrospective nationwide population-based study. Graefe’s Archive for Clinical and Experimental Ophthalmology, 1–11. https://doi.org/10.1007/s00417-021-05093-4
Author Response
Reviewer #3:
I am very glad the authors wrote this paper. It is an accurately, needed and useful for the detection of the progression of AMD. This is a well written paper demonstrating efficacy of visual acuity control in the advancement of AMD.
Specific comments for revision:
Introduction
Comment 1: They should describe in more detail the development of the telemonitoring strategy. Could benefit from some further explanation for clarity.
Clarification: The requested information was added to the introduction.
Change: “The system uses preferential hyperacuity perimetry to detect minute differences in the relative spatial localization of two or more objects and was previously described(1). In brief, during the test of each eye, the patient is responding to fast stimuli in random locations in the visual field of the central 14°. These flashing signals include artificial distortions with varying amplitudes. The marking of the patient in the location of the presented distortion, in a different location, or the absence of a response are collected, transmitted to the secured cloud location, and are analyzed by the system’s artificial intelligence algorithm. Upon identification of a statistically significant change in the testing results compared to a baseline period, a change alert is communicated to the prescribing physician through a remote diagnostic clinic that provides the monitoring service”.
Comment 2: They must include the author (Zapata et al., 2021)
Clarification: The requested reference was added.
Change: AMD has been long established as a common condition in the elderly population (37,38,39).
Materials and Methods
Comment 3: The study deals with the measurement of visual acuity thus, the methodically crucial factor.
In the methods, a number of unclear circumstances can be found which have to be carefully clarified.
For example, type of test in the measurement, light conditions of the room, visual acuity was measured monocularly and binocularly.
Clarification: The visual acuity was measured in the physicians’ offices during routine care visits which were pre-scheduled or triggered by the system alerts. Since this is a report of real-world data, no specific protocol for the measurement of the visual acuity was implemented. Clarification was added to the Methods section. It can be assumed that visual acuity in retina clinics was measured monocularly.
Change: They also included the VA at baseline and confirmed conversion (when available) as measured at the physician office
Comment 4: What were the inclusion and exclusion criteria?
Clarification: The inclusion criteria was added to the Patients and Methods section. No formal exclusion criteria were requested.
Change: The inclusion criteria for enrollment into the program is diagnosis of intermediate dry AMD and best corrected visual acuity of 20/60 or better in any eye that is prescribed.
Comment 5: We have previous evidence suggesting that refractive error may be associated with AMD.
Was the refractive error taken into account?
Clarification: The refractive error was not taken in account, only the best corrected visual acuity and diagnosis of dry AMD.
Comment 6: Diabetes is a risk factor for AMD, stronger for late AMD than for earlier stages. Was it taken into account whether the patients included were diabetic?
Clarification: The inclusion criteria was limited to diagnosis of dry AMD, without specific requirements related to any underlying risk factor.
Results
Comment 7: The tables present the data nicely but a bit more explanation of the content and implications of the data in the tables would be beneficial.
Clarification: Text was added in two places in the results section.
Changes:
Added a missing text in one of the column headers of the table. “VA at Conversion with known baseline VA”.
Added: “Since this report is based on real-world, retrospectively collected data, not all visual acuity values from both time points were available. There was a statistically significant difference in visual acuity between baseline and at the time of conversion (p=0.00014)”.
Discussion
Comment 8: The discussion of data citation was good, but I would have liked to see some views on a way forward. Finally, is that there is what is the practical impact of this study?
Clarification: Added a sentence at the end of the discussion.
Change: “It may drive physicians, patients, and their families to request access, gain benefit from the expected performance and drive a large-scale improvement in the quality of life of this growing population”.

Round 2
Reviewer 2 Report
Thanks for updating, and appropriate for this AMD focussed issue.
Reviewer 3 Report
Comments solved
This manuscript is a resubmission of an earlier submission. The following is a list of the peer review reports and author responses from that submission.